# SDN-Based Intrusion Detection System for Early Detection and Mitigation of DDoS Attacks

**Pedro Manso [1], José Moura [2,*] and Carlos Serrão [3]** 

[1] Department of Information Science and Technology, School of Technology and Architecture, ISCTE—Instituto Universitário de Lisboa, 1649-026 Lisbon, Portugal; pedro_caetano_manso@iscte-iul.pt

[2] Instituto de Telecomunicações (IT), ISCTE—Instituto Universitário de Lisboa, 1649-026 Lisbon, Portugal

[3] Information Sciences, Technologies and Architecture Research Center (ISTAR-IUL), ISCTE—Instituto Universitário de Lisboa, 1649-026 Lisbon, Portugal; carlos.serrao@iscte-iul.pt

[*] Correspondence: jose.moura@iscte-iul.pt

**Abstract:** The current paper addresses relevant network security vulnerabilities introduced by network devices within the emerging paradigm of Internet of Things (IoT) as well as the urgent need to mitigate the negative effects of some types of Distributed Denial of Service (DDoS) attacks that try to explore those security weaknesses. We design and implement a Software-Defined Intrusion Detection System (IDS) that reactively impairs the attacks at its origin, ensuring the "normal operation" of the network infrastructure. Our proposal includes an IDS that automatically detects several DDoS attacks, and then as an attack is detected, it notifies a Software Defined Networking (SDN) controller. The current proposal also downloads some convenient traffic forwarding decisions from the SDN controller to network devices. The evaluation results suggest that our proposal timely detects several types of cyber-attacks based on DDoS, mitigates their negative impacts on the network performance, and ensures the correct data delivery of normal traffic. Our work sheds light on the programming relevance over an abstracted view of the network infrastructure to timely detect a Botnet exploitation, mitigate malicious traffic at its source, and protect benign traffic.

**Keywords:** SDN; DDoS; IDS; mirroring; OpenFlow; botnet

## 1. Introduction

According to [1], in the beginning of 2018, more than 50% of the world population used the Internet (85% in Europe and 95% in North America). The number of devices connected to the Internet has skyrocketed during the last decade [2]. Cisco [3] forecasts that in 2021 there will be around 27.1 billion network devices worldwide. This exponential growth represents not only the progress of technology, however also the opportunity for attackers to take advantage of this extraordinary infrastructure to compromise many network resources and information assets.

In 2016, the world witnessed one of the world's largest Distributed Denial of Service (DDoS) attacks ever seen [4]. This attack was possible due to security weaknesses that allowed the non-authorized remote access to Internet of Things (IoT) devices. In this way, non-identified attackers have installed a Botnet (i.e., Mirai) at a very high number of IoT devices. These devices were located at network domains geographically separated from each other. Then, at a specific instant of time, through the botnet, the compromised IoT devices simultaneously generated a high amount of traffic towards specific Internet Servers, exhausting their resources. Consequently, the services normally provided by those servers were down for several hours due to the huge difficulty to neutralize every attack source.

Software Defined Networking (SDN) is a promising paradigm that allows the programming of the logic behind the network's operation with some abstraction level from the underlying networking devices. Large companies, such as Google, already deploy SDN with great success [5].

Our work proposes a feasible SDN-based generic solution to mitigate modern Mirai-like DDoS attacks when and where they originate. In this way, we obtain a more accurate and flexible detection method without adding too much overload on the network. The current proposal aims to detect and mitigate the DDoS attack at its origin while maintaining the Quality of Service (QoS) of benign users even when an online service is under attack. Moreover, the solution is open and scalable enough to accommodate the detection and mitigation of other types of DDoS attacks at their origin. In a nutshell, our proposed solution has the following characteristics: (i) it compares at runtime the expected trend of normal traffic against the trend of monitored traffic; (ii) if a significant deviation on the traffic trend is detected, then an event is created; (iii) as an event associated to a DDoS attack is produced, then a SDN controller creates flow rules for blocking the malign traffic; and (iv) we assume that the detection and mitigation of a DDoS attack is made at each potential source of that DDoS attack.

This article is organized into five different sections. The initial section introduces the motivation and the goals of our current research. The second section discusses some related work. The following section is devoted to the design, specification, and deployment of our proposal. The fourth section describes the proposal evaluation tests and discusses their results. Finally, the last section discusses the major conclusions and some guidelines for future research.

## 2. Related Work

Cyber-attacks, especially those based on DDoS, are more and more prevalent, and their impact is greater than ever on the network infrastructure, online services, and digital information assets. In parallel, SDN is starting to emerge and is gaining increasing attention from the diverse networking players. We envision that it is very important to propose SDN-based solutions for urgently thwarting DDoS attacks, among others.

There is already a plethora of solutions capable of detecting and fighting against DDoS attacks. To analyze such solutions, we have classified them into two different categories: signature-based and anomaly-based solutions. Signature-based solutions identify each DDoS attack through its signature. The signatures of DDoS attacks are stored in a database. Avant-Guard [6] is one of the most referenced works in this category. It uses two types of modules: Connection Migration (CM) and Actuation Trigger (AT) module. The CM works as a specialized proxy that receives and classifies TCP-SYN requests. Additionally, the AT module generates an event to a system's controller after the CM has classified some traffic as non-legit. Nevertheless, this work just deals with SYN flood attacks. Other literature solution proposes DDoS detection by controlling the bandwidth in each router's interface [7]. After this solution detects some congestion, it notifies the controller and changes the available bandwidth. However, it does not distinguish benign from malign traffic. Other works suggest the integration of an Intrusion Detection System (IDS) with SDN [8]. Further similar signature-based techniques are detailed in [9–16]. Nonetheless, these have serious limitations because they cannot detect novel attacks or known attack variants. The next paragraph debates a more flexible alternative.

The solutions classified as anomaly-based learn about the "normal" behavior of the network. In this way, everything that drives away from the "normal" behavior is classified and reported as a strange event. For instance, the authors of [17] use a neural network for analyzing the network flows. In this way, they can classify each flow as either a normal one or associated to an ongoing DDoS attack. Other similar proposals are [18–20]. However, all these proposals oblige to considerable supervised training before efficiently detecting attacks. A survey of more network anomaly detection techniques is available in [21].

According to our best current knowledge, no previous literature contributions mitigate DDoS-based cyber-attacks directly at their origin, except for some approaches based on clustering [22]. A clustering method is a non-supervised data mining technique. Nevertheless, the clustering method

shows some drawbacks, such as the efficient management of clusters. This can be a very complex task, demanding enormous processing effort, which creates serious system bottlenecks as well as inaccurate decisions [22]. To overcome these potential issues of clustering methods, we alternatively propose an anomaly-based and event-triggered scalable solution to detect these types of cyber-attacks at their source network domains, completely aligned with [23], minimizing the attacks' amplification. Our SDN-based solution mitigates the attack via a blacklist of IP addresses associated with the detected malign flows. Then, these flows are blocked at their source domains using OpenFlow rules. In addition, our anomaly-based detection model uses a threshold value to trigger a remediation process for the detected DDoS-based cyber-attack. The next section discusses the design and implementation of our proposed DDoS mitigation system.

## 3. Design and Deployment of Our Flow-Based Proposal Thwarting DDoS Attacks

This section discusses the design and implementation of our flow-based proposal that intends to detect and mitigate DDoS attacks at their source domains. In addition, our solution protects some relevant Quality of Service (QoS) metrics for the flows that are associated with normal Web Services.

### 3.1. System Design

The proposed system uses a DDoS attack detection and mitigation mechanism that integrates an Intrusion Detection System (IDS) within the SDN architecture at the client side for either domestic or organizational networking scenarios.

The system operates via a loop control among three basic architectural components (Figure 1): the network, the IDS, and the controller. The network represents all the data traffic and where a potentially DDoS attack might be launched. The IDS represents our DDoS attack detection mechanism, which analyzes all the traffic exchanged across the network. As the IDS detects an ongoing DDoS attack, the IDS notifies the Controller. After the Controller is notified by the IDS, the Controller transfers to the networking devices of the data path of some new flow rules for restoring the normal operation of the network as quickly as possible.

The system has three critical phases: detection, communication, and mitigation. The detection phase is about the system's capability of detecting a DDoS attack. The communication phase occurs when the IDS alerts the controller about the detected DDoS attack. The mitigation phase is when the controller transfers some flow rules to the local switch, blocking the evil traffic. These flow rules are stored in a permanent way within that switch.

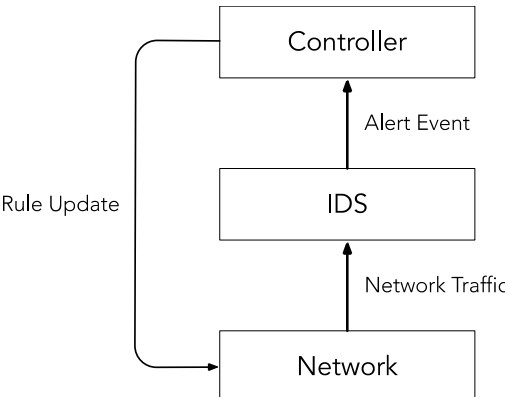

**Figure 1.** The System's Basic Architecture.

Figure 2 shows the conceptual model of the current work. Our proposal combines the functionality offered by both an SDN controller and an Intrusion Detection System (IDS). By aggregating these two entities, we have obtained an SDN-Based IDS Monitor. Each new Packet that arrives to the system (i.e., it was received in the switch's port) is classified as belonging to a Flow and makes a Request to

the system. In this way, the switch processes the received Packet according to a Rule associated to the Flow that Packet belongs to, making a Decision. In the case, the switch does not initially have any Rule for that Flow, then, the switch requests it to the SDN controller. The SDN controller installs the correct flow Rule in the switch which permits the Packet to proceed to its destination only if the packet belongs to a "well behaved" flow. Otherwise, a dropping Rule is installed in the switch. In this last case, the packets of the "bad" flow are discarded. The flow classification as "good" or "bad" is made by the IDS using a set of pre-configured rules. Each Request that is involved in a Flow results in a Decision from the developed system.

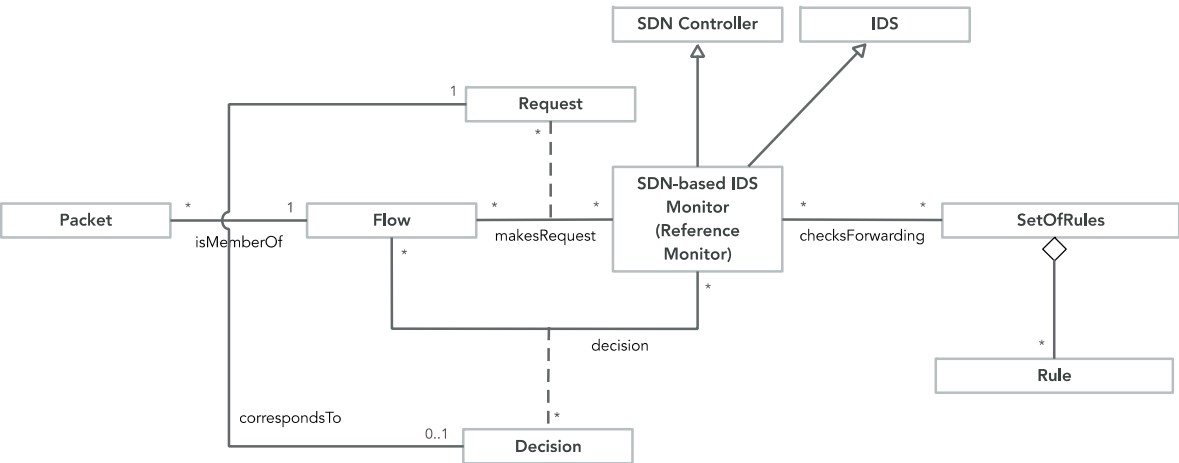

**Figure 2.** The System's Conceptual Model (adapted from [24]).

### 3.2. System Deployment

We have implemented the virtualized system that is visualized in Figure 3 as a proof-of-concept of the system's model that was already presented in Figure 2. Our system uses three different Virtual Machines (VM): VM A, B, and C. VM A contains two modules: the SDN Ryu controller and the Snort Intrusion Detection System (IDS). The SDN Ryu controller programs the network operation when specific flow-based events occur (e.g., after the controller has received an OpenFlow *Packet_in* message). The SDN controller changes the network data plane via link *(A)*, sending OpenFlow forwarding rules. The Snort IDS is our rule-based system is used to "fire an alarm" in the presence of an occurring DDoS attack. Hence, after the IDS detects a DDoS attack, it sends an alert packet via the Unix Domain Socket to the SDN controller, which is visualized in Figure 3 as link *(D)*. Therefore, VM A has two network interfaces. The first interface (adapter 1) was configured as Host-Only (i.e., virtualized networking mode provided by the Hypervisor—VirtualBox) and it supports link *(A)* communication. The second interface (adapter 2) was configured as the internal network (*intnet* mode of the Hypervisor) and it supports link *(B)* communication.

VM B emulates the network domain (e.g., home network) where a potential cyber-attack can be initiated. It runs a network emulator (i.e., Mininet) which deploys a network domain with some hosts, the software-based switch, and the NAT routing device. The software-based switch runs the OpenFlow protocol client part and belongs to the data plane of our testbed. This switch performs port mirroring and sends the entire traffic to the Snort IDS via link *(B)*. The NAT device is used as a gateway for the hosts having access to the online server through link *(C)*.

VM C represents an online server that can be potentially attacked by a DDoS threat. This VM has a single network adapter. This adapter was configured as Host-Only and it supports link *(C)* communication. In a nutshell, our testbed connections are as follows:

- Bi-directional link (A) between the OpenFlow switch and the SDN controller which exchanges control traffic between the data and control planes—the Southbound Application Programming Interface (API);

- Unidirectional link (B) between OpenFlow switch and Snort IDS which enables the switch to send all the mirrored traffic to the IDS for further analysis;
- Bi-directional link (C) which connects the local network domain (i.e., VMs A, B) to the remote service;
- Unidirectional link (D) which enables Snort IDS to notify the SDN controller (i.e., Ryu) about an ongoing DDoS attack through *alert packets* via the Unix Domain Socket.

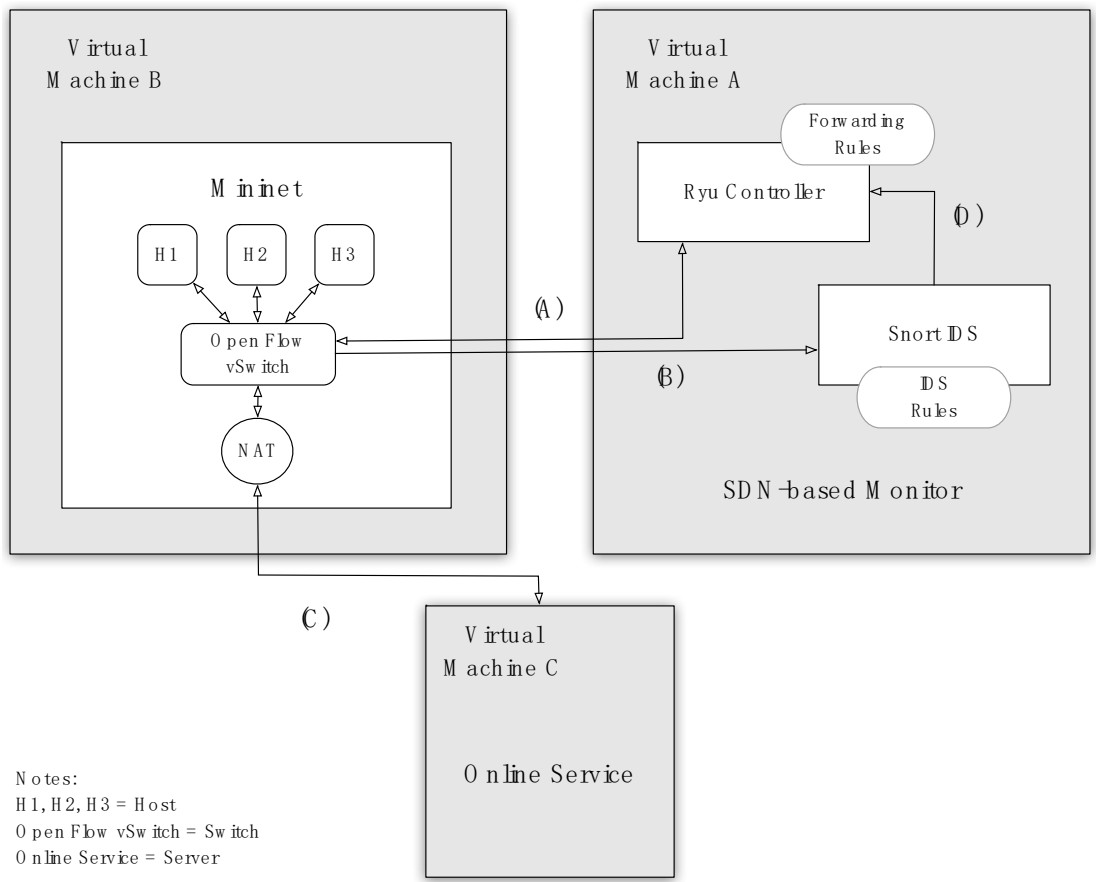

**Figure 3.** System Deployment.

Figure 4 represents the workflow of our system for each packet that arrives to the OpenFlow switch. The reader should now consider a usage scenario where a host tries to send a packet to the online resource. The OpenFlow switch receives that packet and tries to match it against the flow rules of its table. If no match occurs, then the switch requests to the controller a new rule for the new flow. Then, the controller responds by sending the new forwarding rule for that flow. Alternatively, if a match occurs within the switch, this means the switch already has a flow rule for that received packet. In this case, the switch forwards that packet according to the existing flow rule. In this way, the packet traverses the NAT gateway and proceeds towards the online service. In either case, the switch will mirror every received packet to Snort IDS. Then, the IDS analyzes the packet, processing it by means of a statistical function. This is designated as anomaly detection. In this specific case, the statistical function evaluates the mirrored packet as part of a malign flow. Then, the IDS notifies the SDN controller about this. By its turn, the SDN controller sends a blocking flow rule to the switch. The switch can mitigate the attack by eliminating all packets, matching the new installed blocking rule.

Figure 5 gives further details about how our solution operates in the presence of a DDoS attack driven by User Datagram Protocol (UDP) flooding and originated in a compromised internal host. We assume that the SDN controller has already installed a rule in the switch that normally forwards packets that originated from that host. We now explain the diverse processing steps a packet follows

within our proposal. First, the `ICMP Echo Request` arrives to the switch. Then, the switch forwards the `ICMP Echo Request` to NAT. The switch mirrors with some latency (this depends on the internal switch fabric) the `ICMP Echo Request` to the Snort IDS. While NAT forwards the `ICMP Echo Request` to the Server and waits for the `ICMP Echo Reply`, the Snort IDS in parallel analyzes the `ICMP Echo Request` by means of a statistical function. If Snort considers that the processed packet has a statistically incorrect behavior, then the source node of that packet is classified as a malign host. Then, the IDS notifies the SDN controller about that via the *Unix Domain Socket*. After this, the controller mitigates the attack by sending a rule to the switch that blocks future packets that originated from the discovered malign host. Hence, after receiving the previous dropping packet rule, the switch can protect the network resources against the malign packets as well as protecting the remote server from that attack. Nevertheless, the switch suffers from a slight processing overhead due to the *mirroring* function. However, it is worth noting that this overhead is not so relevant in our domestic scenario.

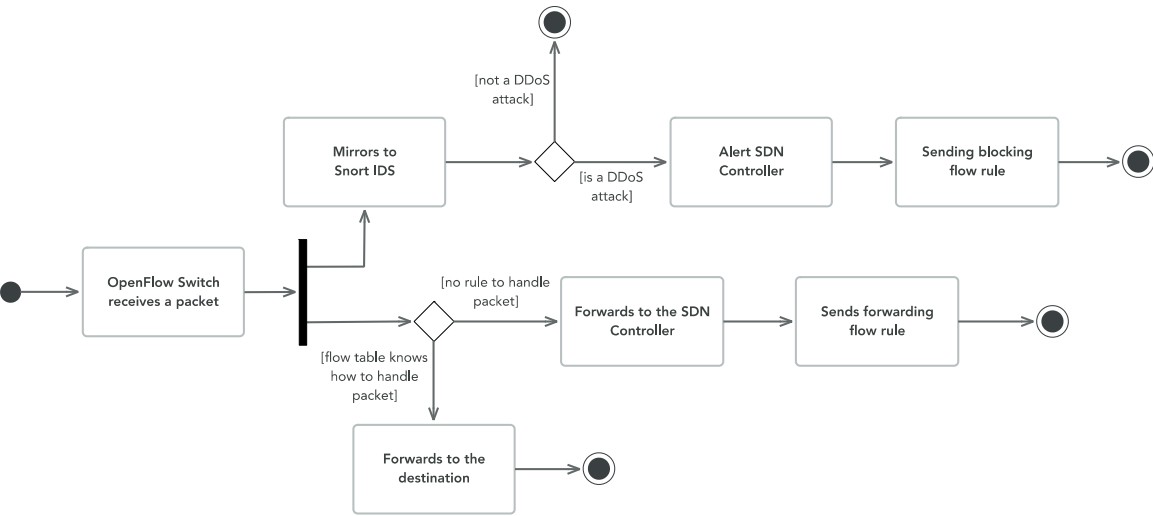

**Figure 4.** System Workflow for Each Received Packet.

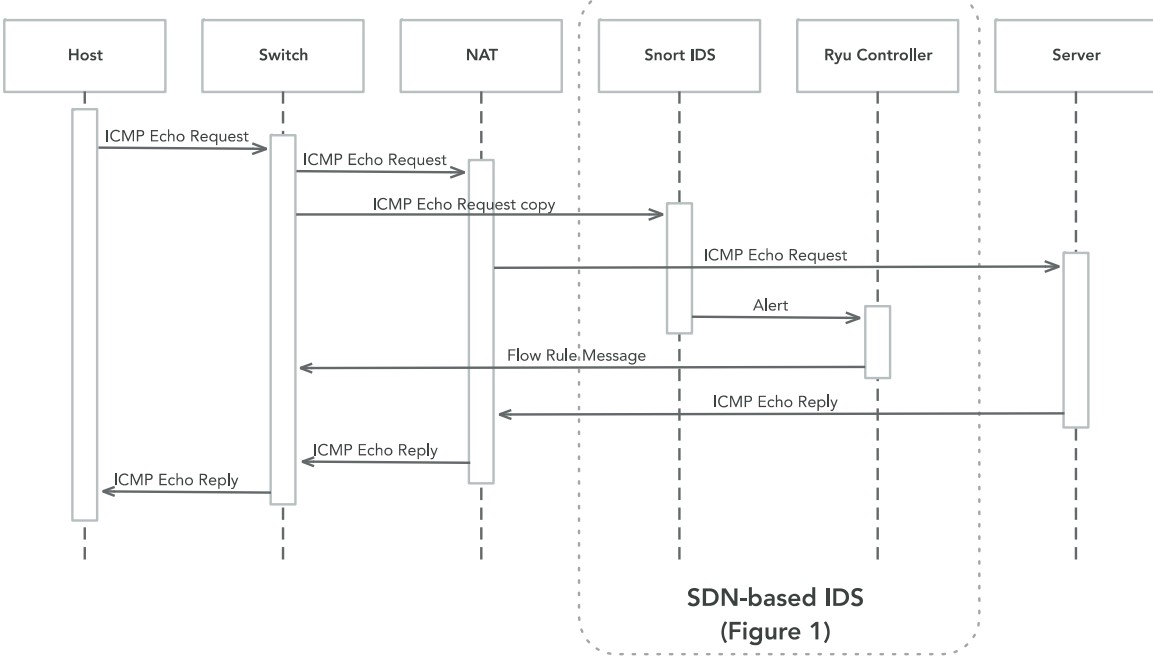

**Figure 5.** Sequential Diagram of System Messages.

To deploy our SDN-based IDS (see Figure 5), we have aggregated the offered functionalities of an SDN controller and an IDS. The SDN controller is based on the open-source Python solution designated as Ryu [25]. We have coded within this controller a Python function designated as `process_snort_alert`. This function processes alerts received from the IDS. If the received alert is detected as a "Ryu block", then that function sends a blocking flow rule to the Switch to otherwise ignore it. We have also used a well-known IDS, Snort [26]. It detects attacks based on known rule signatures. We have configured Snort by editing the files *snort.conf* as well as *ddos_detection.rules* to consider our specific DDoS rules. These DDoS rules are used along the distinct testing scenarios of our proposal and allow the flexibility of our solution to adapt to other DDoS cyber-attacks.

The Mininet [27] was used in our work to emulate the home network. This network was deployed using the Mininet Python API. For that, we created a Python script file. Inside that file, there are some instruction lines for deploying a NAT device within our topology. This NAT device works as a gateway between our emulated home network and the online remote service. The home network IP address is *10.0.0.0/8*. The online service IP address is *192.168.56.104*.

Our proposal uses *Port Mirroring* for traffic monitoring. **Port Mirroring** consists of duplicating packets that go in/out one switch's port and forwards these packets to another switch's port. To configure *port mirroring*, we started by removing the IP address from the interface *enp0s8* of the VM B (sudo `ifconfig enp0s8 0`). Then, we used the *ovs-vsctl* tool to add to the switch *s1* a new port that connects to the interface enp0s8 (sudo `ovs-vsctl add-port s1 enp0s8`). Next, we created a mirror, added it to the switch *s1*, and finally configured it following the instructions from the OpenvSwitch Frequently Asked Questions (FAQ) web site (http://docs.openvswitch.org/en/latest/faq/configuration/?highlight=mirror). The next section discusses the evaluation results of our current proposal.

## 4. Results

This section describes the evaluation tests of our proposal and discusses the obtained results. It also contains useful information about the home simulated network. To simulate the DDoS attack, we used the `hping3` tool that sends customized TCP/IP packets. Table 1 shows the `hping3` arguments that were used throughout our testing scenarios.

**Table 1.** Hping3 Tool Arguments Used in Our Testing Scenarios.

| Arguments | Description |
| --- | --- |
| `-i` | Indicates the interface to use |
| `-flood` | Sends packets as fast as possible without taking care to show incoming replies |
| `-1` | Uses ICMP mode |
| `-a` | Sets a fake IP address |
| `-d` | Defines the size of each packet |

### 4.1. Tests' Description

We now introduce the three scenarios that were used to assess our proposal. Table 2 summarizes these scenarios. We recall that these three scenarios assume that each DDoS attack should be preferably mitigated at its original source domains. In this way, we avoid that the remote servers could suffer the negative performance effects induced by a DDoS attack. During the next discussion, we evaluate if the main ideas behind our proposal are completely fulfilled. Therefore, it is not our intention to perform tests with scenarios with thousands of nodes, because in doing that, a testbed would not be a convenient option. For that, a network simulator would be a more convenient tool. Nevertheless, this option is out of the scope of the current work. In this way, the tests listed in Table 2 have been made with the intent of validating the most relevant functional aspects of our proposal.

**Table 2.** Test Scenarios Summary.

| Scenario | Description | Parameters Evaluated | Hping3 Arguments |
|---|---|---|---|
| I | A Distributed Denial of Service (DDoS) attack is simulated with two malign hosts while a benign host has its normal access to the online service. | DDoS Mitigation Time; Average RTT (Round Trip Time); Packet loss. | `-flood` |
| II | A DDoS attack with a spoofed IP address is simulated with one malign host while two benign hosts have their normal access to the online service. | DDoS Mitigation Time; Average RTT; Packet loss. | `-a;` `-flood` |
| III | A DDoS attack with packet's size manipulation is simulated with one malign host while two benign hosts have their normal access to the online service. | DDoS Mitigation Time; Average RTT; Packet loss. | `-d;` `-flood` |

### 4.2. Results Presentation and Discussion

We now present and discuss the obtained evaluation results. After some packets were exchanged within the network, using the *pingall* Mininet command, the switch updated its flow rule table, as shown in Figure 6. We can verify that new rules were updated within the switch's Flow Table with *priority=1*. Using these flow rules, the switch has the normal behavior of a legacy switch that learns the MAC address of each host and binds it to the switch's port that has last received a frame from that host.

```
NXST_FLOW reply (xid=0x4):
 cookie=0x0, duration=76.891s, table=0, n_packets=3, n_bytes=238, idle_age=71,
priority=1,in_port=2,dl_dst=b2:2a:30:3a:e7:f2 actions=output:1
 cookie=0x0, duration=76.889s, table=0, n_packets=2, n_bytes=140, idle_age=71,
priority=1,in_port=1,dl_dst=00:00:00:00:00:01 actions=output:2
 cookie=0x0, duration=76.877s, table=0, n_packets=3, n_bytes=238, idle_age=71,
priority=1,in_port=3,dl_dst=b2:2a:30:3a:e7:f2 actions=output:1
 cookie=0x0, duration=76.875s, table=0, n_packets=2, n_bytes=140, idle_age=71,
priority=1,in_port=1,dl_dst=00:00:00:00:00:02 actions=output:3
 cookie=0x0, duration=76.870s, table=0, n_packets=3, n_bytes=238, idle_age=71,
priority=1,in_port=4,dl_dst=b2:2a:30:3a:e7:f2 actions=output:1
 cookie=0x0, duration=76.867s, table=0, n_packets=2, n_bytes=140, idle_age=71,
priority=1,in_port=1,dl_dst=00:00:00:00:00:03 actions=output:4
 cookie=0x0, duration=76.858s, table=0, n_packets=3, n_bytes=238, idle_age=71,
priority=1,in_port=3,dl_dst=00:00:00:00:00:01 actions=output:2
 cookie=0x0, duration=76.856s, table=0, n_packets=2, n_bytes=140, idle_age=71,
priority=1,in_port=2,dl_dst=00:00:00:00:00:02 actions=output:3
 cookie=0x0, duration=76.849s, table=0, n_packets=3, n_bytes=238, idle_age=71,
priority=1,in_port=4,dl_dst=00:00:00:00:00:01 actions=output:2
 cookie=0x0, duration=76.847s, table=0, n_packets=2, n_bytes=140, idle_age=71,
priority=1,in_port=2,dl_dst=00:00:00:00:00:03 actions=output:4
 cookie=0x0, duration=76.835s, table=0, n_packets=3, n_bytes=238, idle_age=71,
priority=1,in_port=4,dl_dst=00:00:00:00:00:02 actions=output:3
 cookie=0x0, duration=76.830s, table=0, n_packets=2, n_bytes=140, idle_age=71,
priority=1,in_port=3,dl_dst=00:00:00:00:00:03 actions=output:4
 cookie=0x0, duration=138.890s, table=0, n_packets=97, n_bytes=10017, idle_age=5, priority=0
actions=CONTROLLER:65535
```

**Figure 6.** OpenFlow Switch Flow Rule Table Before a DDoS Attack.

Figure 7 visualizes the typical traffic trend of our network when all hosts are well behaved.

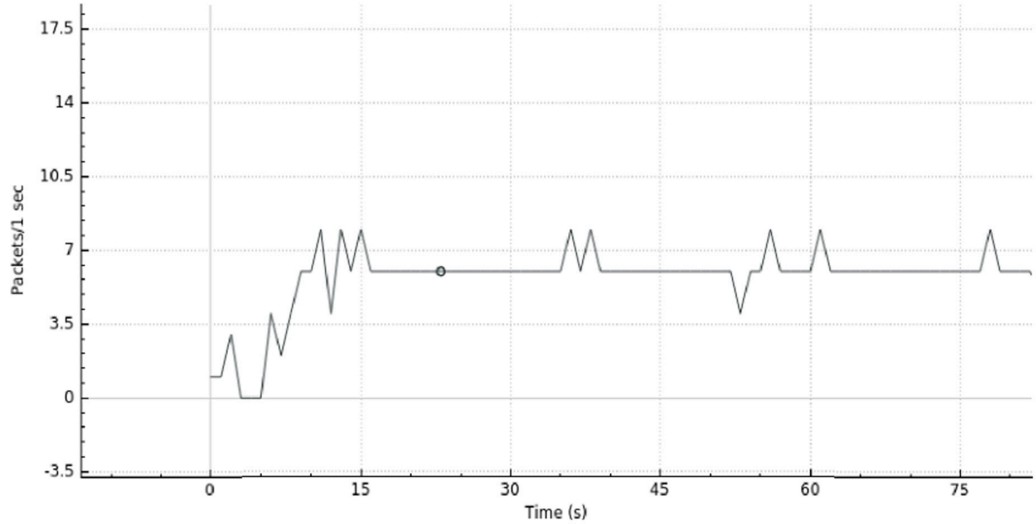

**Figure 7.** Usual System Usage before a DDoS Attack.

Table 3 gives a numerical perspective of the typical values of some *Quality of Service* (QoS) metrics when there is no attack. These results were obtained from *ping* tests that were executed in each host.

**Table 3.** Typical System Performance without DDoS Attack.

| | Host 1 | | Host 2 | | Host 3 | |
|---|---|---|---|---|---|---|
| Test Id. | Average RTT (ms) | Packet Loss (%) | Average RTT (ms) | Packet Loss (%) | Average RTT (ms) | Packet Loss (%) |
| 1 | 0.588 | 0 | 0.439 | 0 | 0.765 | 0 |
| 2 | 0.756 | 0 | 1.274 | 0 | 0.712 | 0 |
| 3 | 0.493 | 0 | 0.463 | 0 | 0.593 | 0 |
| 4 | 0.766 | 0 | 0.904 | 0 | 1.37 | 0 |
| 5 | 2.004 | 0 | 0.402 | 0 | 0.447 | 0 |
| Average | 0.922 | 0 | 0.696 | 0 | 0.777 | 0 |

Figure 8 shows the traffic volume produced by a DDoS Attack when our proposed Defense System is disabled. From this, we can analyze that after 50 s, the network was submitted to a significant amount of traffic, originated by that DDoS attack. In fact, that attack originated a maximum peak that slightly overlapped the value of 42,000 packets/s (2,520,000 packets/m). The current attack is inspired in the Center for Applied Internet Data Analysis (https://www.caida.org/home/ (verified in 27/02/2019)) (CAIDA) dataset which was commonly used in recent publications [28]. In the next sub-sections, we present and discuss the several tests we made to detect and mitigate the diverse cyber-attacks.

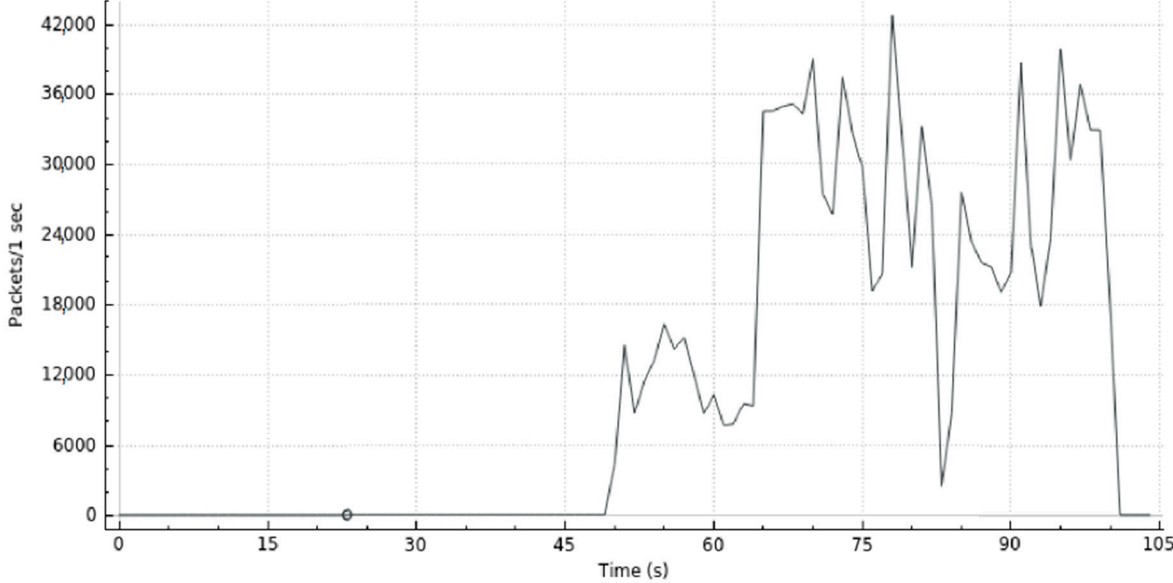

**Figure 8.** Network Traffic injected by a DDoS Attack when our Defense System is Off.

### 4.2.1. Scenario I

As described previously, this test consists of performing a DDoS attack originating in both `hosts 1` and **3**, while `host 2` maintains its "normal" access to the remote server. Therefore, the `hping3` tool was used in ICMP mode, flooding packets as fast as possible to the server (`192.168.56.104`). The `host 2` flow rate while the server is under attack is displayed in Figure 9. We can see that the `host 2` rate varies along the time. These rate variations result from the online server resource starvation and network congestion, which are both induced by the DDoS attack.

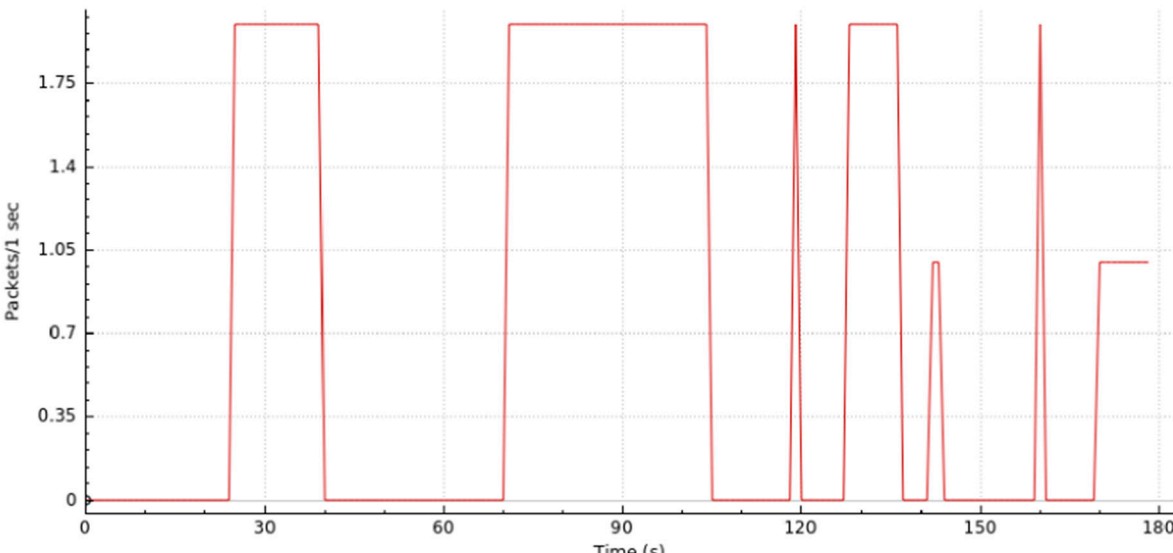

**Figure 9.** Scenario I—Host 2 Rate when the Remote Server is Under a DDoS Attack.

We have configured the Snort to detect a cyber-attack through the rule specified in *(1)*. According to that rule, an alert is generated when Snort captures more than 10 packets within one second from any source IP within the 10.0.0.0/8 network, destined to 192.168.56.104 (i.e., Online web server), and using any Transport ports.

$$\texttt{alert icmp 10.0.0.0/8 any} \rightarrow \texttt{192.168.56.104 any (msg:"ryu block";}$$
$$\texttt{detection\_filter:track by\_src, count 10, seconds 1, sid:1000001)}$$

(1)

When the system detects an attack, it starts the mitigation phase. To mitigate the attack, our proposal transfers some flow rules to the local switch (Figure 10). In fact, one can check that our proposal reacts to the attack through the initial two rules displayed at the head of the flow table of the local switch. These two new rules with high priority values drop the received packets from both malicious users, host 1 and host 3.

```
NXST_FLOW reply (xid=0x4):
 cookie=0x0, duration=25.998s, table=0, n_packets=6370225, n_bytes=267549450, idle_age=105,
priority=100,icmp,dl_src=00:00:00:00:00:01,nw_src=10.0.0.1,nw_dst=192.168.56.104 actions=drop
 cookie=0x0, duration=0.344s, table=0, n_packets=0, n_bytes=0, idle_age=25,
priority=100,icmp,dl_src=00:00:00:00:00:03,nw_src=10.0.0.3,nw_dst=192.168.56.104 actions=drop
 cookie=0x0, duration=900.309s, table=0, n_packets=6, n_bytes=364, idle_age=126,
priority=1,in_port=2,dl_dst=92:64:e9:61:b8:b4 actions=output:1
 cookie=0x0, duration=900.307s, table=0, n_packets=3344, n_bytes=140504, idle_age=126,
priority=1,in_port=1,dl_dst=00:00:00:00:00:01 actions=output:2
```

**Figure 10.** Scenario I—OpenFlow Switch's Flow Table after DDoS Attack.

Our solution only blocks the malicious mirrored traffic after it was detected the first time. Therefore, Snort was configured to limit the logging events for 60 s (Figure 11). In this way, the controller only deals with a single alert within that period, avoiding the controller overload. This configuration to limit the number of alerts was defined in the file *threshold.conf*.

```
event_filter gen_id 0, sig_id 0, type limit, track by_src, count 1, seconds 60
```

**Figure 11.** Snort Extra Configuration.

A graphical representation of the DDoS attack is shown in Figure 12 where we can see the traffic generated as a function of time. Here, one can observe the normal usage of the system during the first 60 s. Then, host 3 launches a DoS attack (first peak of packets transmitted per second). Next, host 1 launches another DoS attack (second peak of packets transmitted per second), simulating the DDoS attack. The DDoS attack was nearly mitigated at 64 seconds. Hence, from then on, we can verify that the network stabilizes around its "normal" operating status.

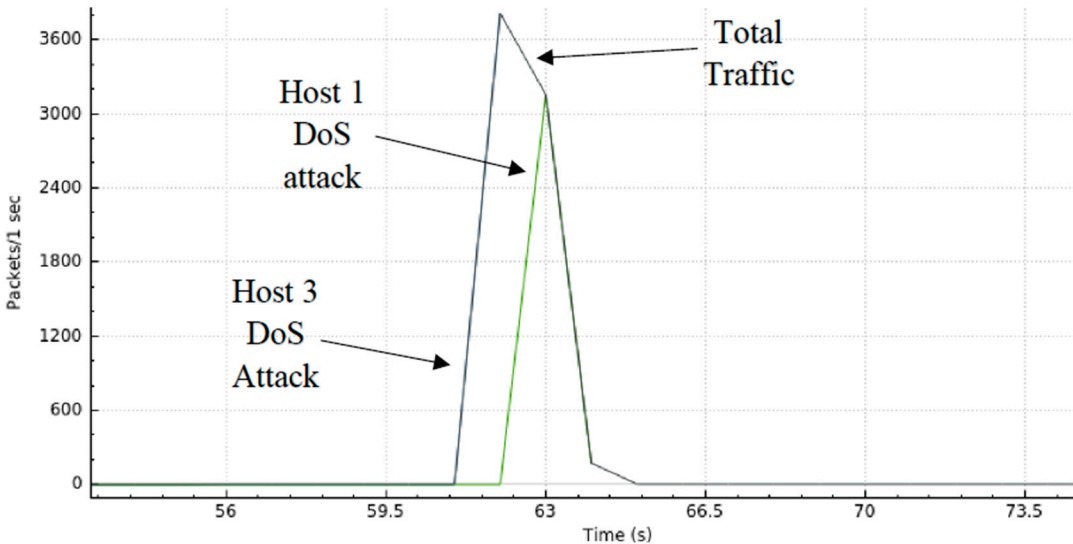

**Figure 12.** Scenario I—DDoS Attack and Mitigation.

Table 4 shows our results to the variables: DDoS mitigation time, the average *Round Trip Time (RTT)*, and the percentage of the packet loss. The results of the average RTT and the percentage of packet loss were retrieved from the *ping* command of host 2. The DDoS mitigation time was retrieved from the Wireshark I/O graph. Our solution needs, on average, approximately five seconds to detect and mitigate the DDoS attack.

**Table 4.** Scenario I—Performance Results.

| # Tests | DDoS Mitigation Time (s) | Average RTT (ms) | Packet Loss (%) |
|---------|--------------------------|------------------|-----------------|
| 1 | 3 | 0.619 | 0 |
| 2 | 4 | 0.547 | 0 |
| 3 | 5 | 0.561 | 0 |
| 4 | 5 | 0.484 | 0 |
| 5 | 7 | 0.767 | 0 |
| Average | 4.8 | 0.596 | 0 |

### 4.2.2. Scenario II

In this scenario, we perform a DDoS attack with a spoofed IP address. Therefore, the hping3 tool was used in ICMP mode, flooding packets as fast as possible to the server IP address (*192.168.56.104*) with a spoofed IP address, i.e., *10.0.0.55*.

To give a better idea of the QoS impact in both hosts 2 and 3 while the network was under attack, we present the results visualized in Figure 13. Even though we simulated the attack with only one host, we can verify that the host's rate remained constant.

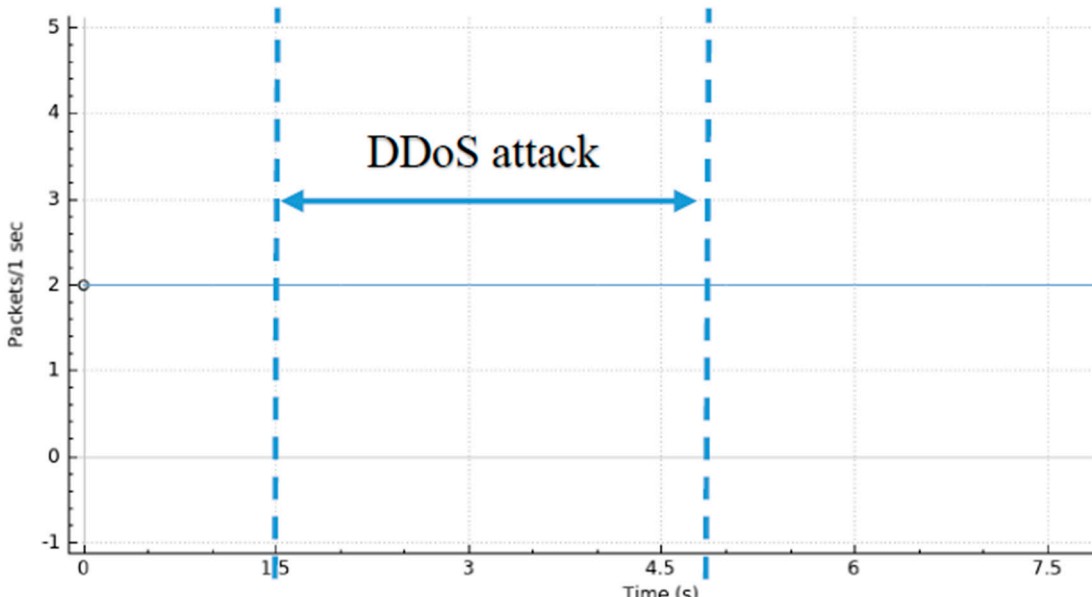

**Figure 13.** Scenario II—Hosts 2 and 3 Rate during a DDoS Attack Dissimulated by Spoofing.

We can confirm that the mitigation process is taking over by checking the switch's flow rule table (Figure 14). We can verify that the controller mitigated the attack through the first two lines of the Table. It consists of a new rule with higher priority to drop packets for the malign host (*priority=100, actions=drop, dl_src=00:00:00:00:00:01, nw_src=10.0.0.55* and *nw_dst=192.168.56.104*).

```
NXST_FLOW reply (xid=0x4):
 cookie=0x0, duration=0.383s, table=0, n_packets=1402250, n_bytes=58894500, idle_age=294,
 priority=100,icmp,dl_src=00:00:00:00:00:01,nw_src=10.0.0.55,nw_dst=192.168.56.104 actions=drop
 cookie=0x0, duration=385.734s, table=0, n_packets=38913, n_bytes=1635522, idle_age=91,
 priority=1,in_port=2,dl_dst=7a:02:17:a7:a9:b2 actions=output:1
 cookie=0x0, duration=385.732s, table=0, n_packets=24, n_bytes=2128, idle_age=91,
```

**Figure 14.** Scenario II—OpenFlow Switch's Flow Table after the DDoS Attack.

The rules used in this test were the same as the test before and the rule specified in *(2)*. The detection filter is the same except for all IP addresses that are not in the 10.0.0.0/8 network. This way, a wider range of IP addresses is under "surveillance".

$$\text{alert icmp !10.0.0.0/8 any}\rightarrow\text{192.168.56.104 any (msg:"ryu block";}$$
$$\text{detection\_filter:trackby\_src,count 10,seconds 1,sid:1000002}$$

(2)

A graphical visualization of the whole process (i.e., detection and mitigation phase) is presented in Figure 15. As can be seen, it illustrates the traffic generated as a function of time. During the initial two seconds, the network operated in its normal status. Then, at second 2, host 3 launched a DDoS attack. This attack was blocked after two seconds from its start. From four seconds until the end of our test, we can see the traffic stabilizing and going back to the "usual" traffic loading.

Table 5 shows our results to the variables: DDoS mitigation time, the average *Round Trip Time (RTT)*, and the percentage of the packet loss. The results of the average RTT and the percentage of packet loss were retrieved from the host 2 ping command. The DDoS mitigation time was retrieved from the Wireshark I/O graph.

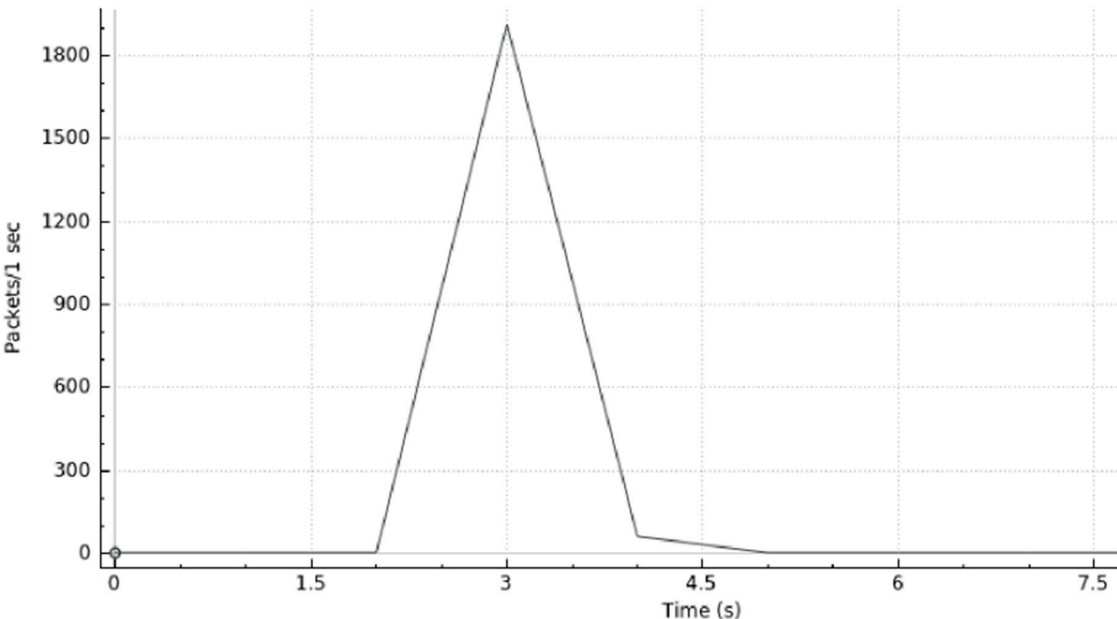

**Figure 15.** Scenario II—DDoS Detection and Mitigation.

**Table 5.** Scenario II—Performance Results.

| # Tests | DDoS Mitigation Time (s) | Average RTT (ms) | Packet Loss (%) |
|---|---|---|---|
| 1 | 3 | 0.504 | 0 |
| 2 | 2 | 0.460 | 0 |
| 3 | 2 | 0.557 | 0 |
| 4 | 2 | 0.513 | 0 |
| 5 | 2 | 0.523 | 0 |
| Average | 2.2 | 0.511 | 0 |

### 4.2.3. Scenario III

The transmission rate of hosts 2 and 3 are presented in Figure 16. We can observe that the rate of these two hosts, despite the cyber-attack occurrence, stayed unchanged during the entire duration of our test. These results show that some QoS characteristics of the normal traffic were protected from the negative effect of the cyber-attack by the collaborative effort of both Snort and the Ryu SDN Controller.

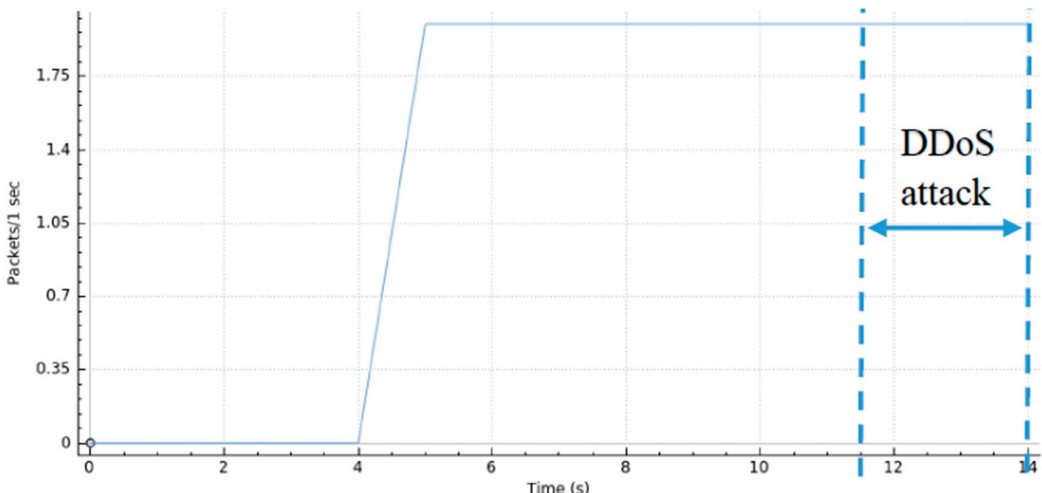

**Figure 16.** Scenario III—Hosts 2 and 3 Rate During a DDoS Attack Based on Large Size Packets.

Through the switch's flow rule table (Figure 17), we can verify that the SDN controller successfully mitigated the attack through the first rule of that table.

```
NXST_FLOW reply (xid=0x4):
  cookie=0x0, duration=69.740s, table=0, n_packets=9049469, n_bytes=8524579542, idle_age=71,
  priority=100,icmp,dl_src=00:00:00:00:00:01,nw_src=10.0.0.1,nw_dst=192.168.56.104 actions=drop
  cookie=0x0, duration=268.081s, table=0, n_packets=19506, n_bytes=18362556, idle_age=79,
  priority=1,in_port=2,dl_dst=5a:2b:35:b6:01:2d actions=output:1
  cookie=0x0, duration=268.080s, table=0, n_packets=4512, n_bytes=4239052, idle_age=79,
  priority=1,in_port=1,dl_dst=00:00:00:00:00:01 actions=output:2
```

**Figure 17.** Scenario III Test OpenFlow Switch's Flow Table after DDoS Attack.

The rule used in this test is in *(3)*. In this case, Snort detects a DDoS attack if a packet that comes from the *10.0.0.0/8* network towards the server IP address has more than 800 bytes of data (*dsize:>800*).

$$\text{alerticmp } 10.0.0.0/8 \text{ any} \rightarrow 192.168.56.104 \text{ any (msg:"ryu block"; dsize: } >800; \text{ sid:1000003)} \quad (3)$$

Next is presented a graphical representation (Figure 18) of the system's load during the detection and mitigation phases. During the initial 11 s, the network shows a normal status. Then, at 11 seconds, **host 1** launched the DoS attack. This attack is mitigated and blocked in nearly 3 s. After 14 seconds, the network traffic stabilizes returning to its "normal" load.

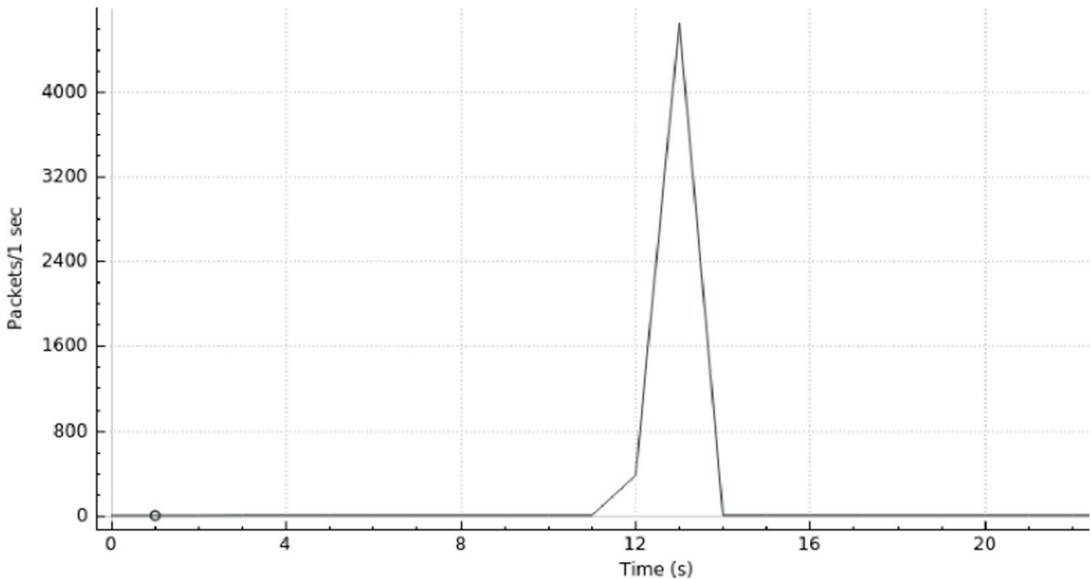

**Figure 18.** Scenario III—DDoS Detection and Mitigation.

Table 6 displays the obtained results according to the DDoS mitigation time, the average *Round Trip Time (RTT)*, and the percentage of the packet loss. The results of the average RTT and the percentage of packet loss were retrieved from the host 2 ping command. The DDoS mitigation time was retrieved from the Wireshark I/O graph.

**Table 6.** Scenario III—Performance Results.

| # Tests | DDoS Mitigation Time (s) | Average RTT (ms) | Packet Loss (%) |
|---------|--------------------------|------------------|-----------------|
| 1 | 3 | 0.674 | 0 |
| 2 | 2 | 0.480 | 0 |
| 3 | 2 | 0.430 | 0 |
| 4 | 2 | 0.449 | 0 |
| 5 | 2 | 0.547 | 0 |
| Average | 2.2 | 0.516 | 0 |

## 5. Conclusions

In this article, we have presented a security system based on the SDN paradigm at the client side to ensure in a centralized way the "normal operation" of a domestic or business networking scenario. Our proposal detects DDoS-based cyber-attack scenarios and limits them at their origin at the client side, this way mitigating the negative consequences of the widespread effect of that attack for potential victims.

For the sake of this manuscript, all the conducted attacks used the UDP protocol in different simple scenarios: a DDoS attack, DDoS attack with IP spoofing, and a DDoS with IP packet size manipulation. The system was able to detect all of them with an average DDoS mitigation time of 3.07 s, an average RTT of 0.541 milliseconds, and without any noticeable packets loss (Table 7). These results suggest that the design and implementation of a security system based on the SDN paradigm, at the client side, can help the mitigation of DDoS attacks while maintaining the normal operation of a network. A relevant learned lesson from the current work is the fact that it is very important to reduce the amount of control events originated by the IDS module and destined to the SDN controller. Otherwise, the SDN controller becomes very congested, taking too much time to discover and mitigate a DDoS attack.

**Table 7.** Comparison among the Results obtained from Our Evaluation Scenarios.

| Scenario | DDoS Mitigation Time (s) | AVG RTT (ms) | Packet Loss (%) |
|---|---|---|---|
| Normal Usage | - | 0.798 | 0 |
| Scenario I | 4.80 | 0.596 | 0 |
| Scenario II | 2.20 | 0.511 | 0 |
| Scenario III | 2.20 | 0.516 | 0 |
| Average | 3.07 | 0.541 | 0 |

The current proposal was developed considering scalability and adaptability to other types of DDoS-based cyber-attacks, with room for improvement, and we consider that adding the capability to "forget" ex-malicious devices is an important feature. This means that a device must not be permanently "blocked" after the occurrence of a DDoS attack was enabled in some degree by that device. We also think that machine learning techniques [29,30] can complement and enhance our rule-based detection mechanism, giving it the capability of detecting new and sophisticated cyber-attacks. Aligned with [31], we also think that the best and effective approach to battle against DDoS attacks is to build a defense mechanism as close as possible to the attack source that generates rogue traffic. This defense mechanism requires collaboration among various service providers to validate the source addresses of packets and deploy other filtering features based on the analysis of flows. This analysis can be performed via a federation of SDN controllers, one for each service provider.

**Author Contributions:** P.M. has contributed with the design, implementation, and evaluation of the current research proposal. J.M. and C.S. have contributed with work supervision and the final proofreading of the current manuscript.

**Funding:** The work of J. Moura was supported by Instituto de Telecomunicações, Lisbon, under Grant UID/EEA/50008/2019.

**Acknowledgments:** Jose Moura acknowledges the support given by Instituto de Telecomunicações, Lisbon. Carlos Serrão acknowledges the support given by Information Sciences, Technologies and Architecture Research Center (ISTAR-IUL).

**Conflicts of Interest:** The authors declare no conflict of interest.

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
