# Peer review of "SDN-Based Intrusion Detection System for Early Detection and Mitigation of DDoS Attacks"

_information, doi:10.3390/info10030106_

Round 1
Reviewer 1 Report
The authors propose a specialized IDS to control DDoS attacks based on the Snort IDS and taking advantage of SDN to thwart the attack.
The idea is interesting and apparently original but their presentation is poor.
Abstract: it says that this system “timely detects several types of cyber-attacks”; however, it only detects DDoS attacks.
System design: The models in this section are imprecise, why not use UML?
The models are also not consistent with each other: the sequence diagram has objects Host, Switch, and Server which do not appear in Fig. 2. In fact, Fig. 2 should be a class diagram with classes corresponding to the objects of Fig. 4. The authors could consult the patterns for IDS presented in [1] which can guide them on this aspect.
Fig. 3 should be a UML activity diagram that allows to describe concurrengt activities, an aspect not clear in their imprecise workflow.
The concepts Snort IDS, Mininet, and Ryu controller should be defined in the introduction or at the beginning of Section 3.
It would be good to separate abstract aspects from product aspects (Mininet, Snort, Ryu).
Section 4. Their experiments use two malign hosts. This is not realistic, DDoS attacks use thousands of attacking hosts. How scalable are these results? A discussion of the applicability of their results is lacking.
The English is poor with many grammatical errors.
[1] E.B.Fernandez, “Security patterns in practice: Building secure architectures using software patterns”, Wiley Series on Software Design Patterns, 2013.
Author Response
Dear Anonymous Reviewer #1,
We thank you for your valuable comments and remarks: these have been very helpful for improving the quality of our manuscript. We have revised the paper as directed by all the comments we received about our previous submission. In this response letter, we quote the original comments received along with our corresponding answers, explaining how the issues raised have been addressed in the revised paper. The reviewers’ comments are in regular font, and our responses are in bold font.
--
The authors propose a specialized IDS to control DDoS attacks based on the Snort IDS and taking advantage of SDN to thwart the attack. The idea is interesting and apparently original but their presentation is poor.
Abstract: it says that this system “timely detects several types of cyberattacks”; however, it only detects DDoS attacks.
R: We have clarified the abstract: “ … automatically detects several DDoS attacks, …”
System design: The models in this section are imprecise, why not use UML? The models are also not consistent with each other: the sequence diagram has objects Host, Switch, and Server which do not appear in Fig. 2. In fact, Fig. 2 should be a class diagram with classes corresponding to the objects of Fig. 4. The authors could consult the patterns for IDS presented in [1] which can guide them on this aspect.
Fig. 3 should be a UML activity diagram that allows to describe concurrengt activities, an aspect not clear in their imprecise workflow.
R: We have significantly worked in Section 3: i) Moved text parts (and Figures) related to the implementation, which were before within the Design, to the correct sub-section, i.e. the one associated to the system’s Design; ii) we have inserted in the Design sub-section a new Figure (Figure 2 of the most recent paper version): it represents by UML the system’s conceptual model of our current work. In this way, we have tried to address the issues raised by the anonymous reviewer.
The concepts Snort IDS, Mininet, and Ryu controller should be defined in the introduction or at the beginning of Section 3.
R: We think that these issues were also addressed with the significant amount of changes we have made in Section3.
It would be good to separate abstract aspects from product aspects (Mininet, Snort, Ryu).
R: We think that these issues were also addressed with, as we already mentioned it, the significant amount of changes we have made in Section3.
Section 4. Their experiments use two malign hosts. This is not realistic, DDoS attacks use thousands of attacking hosts. How scalable are these results? A discussion of the applicability of their results is lacking.
R: We have added in the beginning of sub-section 3.2 (System Deployment) the next phrase: “We have implemented the virtualized system visualized in Figure 3, as a proof-of-concept of the system’s model already presented in Figure 2.”. We have also added in the beginning of sub-section 4.1 (Tests Description) the next text: “We recall these three scenarios assume each DDoS attack should be preferably mitigated at its original source domains. In this way, we avoid that the remote servers could suffer the performance negative effects induced by a DDoS attack. During the next discussion, we evaluate if the main ideas behind our proposal are completely fulfilled. Therefore, it is not our intention to perform tests with scenarios with thousands of nodes because for doing that a testbed would not be a convenient option. For that a network simulator is a more convenient tool. Nevertheless, this option is out of scope of the current work.”
The English is poor with many grammatical errors.
R: The manuscript has undergone a thorough proofreading round.
[1] E.B.Fernandez, “Security patterns in practice: Building secure architectures using software patterns”, Wiley Series on Software Design Patterns, 2013.
--
R: We thank the anonymous reviewer once again for all the valuable feedback on our work. In this revision, we have invested our best efforts to satisfy all the questions and concerns that were raised. The manuscript has undergone a thorough proofreading round, and hopefully it is much improved in terms of clarity and interest to potential readers.
END
Reviewer 2 Report
The proposal introduces a novel SDN-Based Intrusion Detection/Mitigation approach against DDoS. Although the presented work deepens in an emerging problem, the existing state-of-the art has a plethora of similarly solutions, so the authors should stress to highlight the differentiation of their contributions from previous proposals. With this in mind, the following modifications are suggested:
The Introduction must explicitly enumerate the main contributions of the performed research
Thorough the proposal it is implicitly assumed that DDoS threats are based on flooding. ¿Is it also effective against complexity-based attacks, i.e. SlowDrop, Slowloris, etc.?
Design principles like principal/secondary objectives, null/alternative research hypothesis, assumptions/requirements, limitations, etc. should be detailed.
Is the proposal robust against adversarial ML based evasion procedures (e.g. mimicry)?
The “home network” emulation does not seem representative enough of a real monitoring environment. Although the lack of proper and up-to-date datasets is a well-known problem for the research community, this reviewer strongly suggests the validation of the proposal with some functionally standardized datasets for DDoS detectors validation, like KDDcup, CAIDA, etc. In this way it is also compare the obtained results with those of previous publications.
Author Response
Dear Anonymous Reviewer #2,
We thank you for your valuable comments and remarks: these have been very helpful for improving the quality of our manuscript. We have revised the paper as directed by all the comments we received about our previous submission. In this response letter, we quote the original comments received along with our corresponding answers, explaining how the issues raised have been addressed in the revised paper. The reviewers’ comments are in regular font, and our responses are in bold font.
--
The proposal introduces a novel SDN-Based Intrusion Detection/Mitigation approach against DDoS. Although the presented work deepens in an emerging problem, the existing state-of-the art has a plethora of similarly solutions, so the authors should stress to highlight the differentiation of their contributions from previous proposals. With this in mind, the following modifications are suggested:
The Introduction must explicitly enumerate the main contributions of the performed research
R: We have added in the final part of the Introduction the next text: “In a nutshell, our proposed solution has the next characteristics: i) it compares at runtime the expected trend of normal traffic against the trend of monitored traffic; ii) if a significant deviation on the traffic trend is detected, then an event is created; iii) as an event associated to a DDoS attack is produced, then a SDN controller creates flow rules for blocking the malign traffic; iv) we assume that the detection and mitigation of a DDoS attack is made at each potential source of that DDoS attack.”
Thorough the proposal it is implicitly assumed that DDoS threats are based on flooding. ¿Is it also effective against complexity-based attacks, i.e. SlowDrop, Slowloris, etc.?
Design principles like principal/secondary objectives, null/alternative research hypothesis, assumptions/requirements, limitations, etc. should be detailed.
Is the proposal robust against adversarial ML based evasion procedures (e.g. mimicry)?
R: We have inserted in the Introduction the following text: “. Moreover, the solution is open and scalable enough to accommodate the detection and mitigation of other types of DDoS attacks at their origin.” To clarify better how our solution can be expanded, we explain now how we think our solution could defeat the Slowloris attack. For that, we would need to enable an IDS rule for a specific pair of IP addresses (source, destination) that limits the number of ports being allocated in the same server (destination) previously requested by the same client device (source). If that rule is fired then a local event is created, and after that our system will react in a similar way to the one explained within the paper: dropping the logical connections over a specific threshold for a specific pair of IPs (client, server). The overall design goal of our system is to allow its scalability and adaptability to other types of DDoS-based cyber-attacks, simply by the modification of the detection rules. The system was tested with a set of cyber-attacks and reported on this paper, but it maybe expanded to defeat other types of cyber-attacks.
The “home network” emulation does not seem representative enough of a real monitoring environment. Although the lack of proper and up-to-date datasets is a well-known problem for the research community, this reviewer strongly suggests the validation of the proposal with some functionally standardized datasets for DDoS detectors validation, like KDDcup, CAIDA, etc. In this way it is also compare the obtained results with those of previous publications.
R: The “home network” emulation is the main aspect behind our current work – also, it is important to notice that it may also used on other office-like environments. To clarify better this relevant aspect of our work, we have introduced in the beginning of sub-section 4.1 (Tests Description) the next text: “We recall these three scenarios assume each DDoS attack should be preferably mitigated at its original source domains. In this way, we avoid that the remote servers could suffer the performance negative effects induced by a DDoS attack. During the next discussion, we evaluate if the main ideas behind our proposal are completely fulfilled. Therefore, it is not our intention to perform tests with scenarios with thousands of nodes because for doing that a testbed would not be a convenient option. For that a network simulator is a more convenient tool. Nevertheless, this option is out of scope of the current work.” We have also added after Table 3 the next text: “In fact, that attack originated a maximum peak that slightly overlapped the value of 42,000 packets/s (2,520,000 packets/m). The current attack is inspired in the CAIDA dataset that was commonly used in recent publications [27].”. Also, the original goal for the design of this solution was to mitigate Mirai-like bonnets, that compromise IoT devices on the networks (including home and office networks) as a way to prevent the flow of malicious network traffic to potential victims, typically popular Internet servers.
--
R: We thank the anonymous reviewer once again for all the valuable feedback on our work. In this revision, we have invested our best efforts to satisfy all the questions and concerns that were raised. The manuscript has undergone a thorough proofreading round, and hopefully it is much improved in terms of clarity and interest to potential readers.
END
Reviewer 3 Report
The paper suggests an IDS SDN integration solution that aims the mitigation of the DDoS attacks. As a proof of concept the proposed solution is tested with three virtual machines and an emulated home network (mininet).
The key idea is good, it is important to keep the network stable and secure. The introduction is good, the description of possible defenses is complete, both the signature-based and the anomalous solutions are well characterized.
The security system described can be used and its phases are correct. The rules are correct and can be further expanded. The communication is efficient and the mitigation performed by the SDN controller is correct based on the obtained results.
Remarks:
Page 4: "(C) – Bi-directional link that makes the connection between the network devices and an online service." --> the link presented in Figure 2 is uni-directional
Page 4: "Then, the switch forwards the ICMP Echo Request to NAT. At almost the same time, the switch will mirror the ICMP Echo Request to the Snort IDS." --> what does it mean " almost the same time "? What is the latency in case of a collision?
Page 5: "The current proposal connects SDN with Snort IDS for mitigating DDoS attacks. It mitigates each attack at its source and, the location of the Snort IDS at the network edge does not increase the network overhead although it might increase the local switch overhead due to mirroring processing." --> here the key question is what the overhead of the local switch is? More details about this topic and its consequences should be included into the paper.
Author Response
Dear Anonymous Reviewer #3,
We thank you for your valuable comments and remarks: these have been very helpful for improving the quality of our manuscript. We have revised the paper as directed by all the comments we received about our previous submission. In this response letter, we quote the original comments received along with our corresponding answers, explaining how the issues raised have been addressed in the revised paper. The reviewers’ comments are in regular font, and our responses are in bold font.
--
The paper suggests an IDS SDN integration solution that aims the mitigation of the DDoS attacks. As a proof of concept the proposed solution is tested with three virtual machines and an emulated home network (mininet).
The key idea is good, it is important to keep the network stable and secure.
The introduction is good, the description of possible defenses is complete, both the signature-based and the anomalous solutions are well characterized.
The security system described can be used and its phases are correct. The rules are correct and can be further expanded. The communication is efficient and the mitigation performed by the SDN controller is correct based on the obtained results.
Remarks:
Page 4: "(C) – Bi-directional link that makes the connection between the network devices and an online service." --> the link presented in Figure 2 is uni-directional
R: The Figure has been corrected.
Page 4: "Then, the switch forwards the ICMP Echo Request to NAT. At almost the same time, the switch will mirror the ICMP Echo Request to the Snort IDS." --> what does it mean " almost the same time "? What is the latency in case of a collision?
R: We have tried to clarify the issue pointed out by the anonymous reviewer by replacing the previous text with the next one: “The switch mirrors with some latency (that depends on the internal switch fabric) the ICMP Echo Request to the Snort IDS.”.
Page 5: "The current proposal connects SDN with Snort IDS for mitigating DDoS attacks. It mitigates each attack at its source and, the location of the Snort IDS at the network edge does not increase the network overhead although it might increase the local switch overhead due to mirroring processing." --> here the key question is what the overhead of the local switch is? More details about this topic and its consequences should be included into the paper.
R: We have tried to clarify the issue pointed out by the anonymous reviewer by replacing the previous text with the next one: “Hence, the switch after receiving the previous dropping packet rule can protect the network resources against malign packets as well protecting the remote server from that attack. Nevertheless, the switch suffers from a slight processing overhead due to the mirroring function. Noteworthy, this overhead is not so relevant in our domestic scenario.”
--
R: We thank the anonymous reviewer once again for all the valuable feedback on our work. In this revision, we have invested our best efforts to satisfy all the questions and concerns that were raised. The manuscript has undergone a thorough proofreading round, and hopefully it is much improved in terms of clarity and interest to potential readers.
END
Round 2
Reviewer 1 Report
The paper has improved considerably. However, I still have a few objections that should be addressed before publication.
Fig. 2 is based on the Reference Monitor pattern of [1]. However, no credit is given and it appears to have some errors: Why the IDS and the SDN Controller are Reference Monitors? A Reference Monitor enforces authorization rules but an IDS just detects anomalies and may apply blocking rules. In fact, the model shows a set of Authorization Rules, how are they used in the IDS? A packet appears as belonging to many flows, is that possible? UML does not use stars for multiplicities, they should be asterisks. This model needs explanations, it is given without any explanation now.
The Monitor and the rules do not appear in Fig. 3 or its explanation, what happened to them?
[1] E.B.Fernandez, “Security patterns in practice: Building secure architectures using software patterns”, Wiley Series on Software Design Patterns, 2013.
Author Response
We thank you again for your valuable comments and remarks: these have been very helpful for improving the quality of our manuscript.
Fig. 2 is based on the Reference Monitor pattern of [1]. However, no credit is given and it appears to have some errors: Why the IDS and the SDN Controller are Reference Monitors? A Reference Monitor enforces authorization rules but an IDS just detects anomalies and may apply blocking rules. In fact, the model shows a set of Authorization Rules, how are they used in the IDS? A packet appears as belonging to many flows, is that possible? UML does not use stars for multiplicities, they should be asterisks.
R: We have improved Fig. 2 to better reflect our system and to be aligned with the Reference Monitor pattern. We also changed the caption of the figure to reflect the appropriate credits to the original authors. The image was also modified to be aligned with the correct UML notation.
This model needs explanations, it is given without any explanation now.
R: We have included text about Fig. 2 that provides a more detailed explanation of the system.
The Monitor and the rules do not appear in Fig. 3 or its explanation, what happened to them?
R: Modifications on the image have been made to clarify this aspect.
Additionally, we have also improved the readability of the document and improved the English. We have also improved the quality of some of the images on the document.
Reviewer 2 Report
All suggested changes have been successfully completed, so this reviewer considers the submission, ready for publication.
Author Response
We thanks your valuable comments and remarks: these have been very helpful for improving the quality of our manuscript.
Reviewer 3 Report
The authors carried out the suggested modifications.
Author Response

(The authors gave the same response as above.)
